# But How Does It Work in Theory?
# Linear SVM with Random Features

**Yitong Sun**
Department of Mathematics
University of Michigan
Ann Arbor, MI, 48109
syitong@umich.edu

**Anna Gilbert**
Department of Mathematics
University of Michigan
annacg@umich.edu

**Ambuj Tewari**
Department of Statistics
University of Michigan
tewaria@umich.edu

## Abstract

We prove that, under low noise assumptions, the support vector machine with $N \ll m$ random features (RFSVM) can achieve the learning rate faster than $O(1/\sqrt{m})$ on a training set with $m$ samples when an optimized feature map is used. Our work extends the previous fast rate analysis of random features method from least square loss to 0-1 loss. We also show that the reweighted feature selection method, which approximates the optimized feature map, helps improve the performance of RFSVM in experiments on a synthetic data set.

## 1 Introduction

Kernel methods such as kernel support vector machines (KSVMs) have been widely and successfully used in classification tasks (Steinwart and Christmann [2008]). The power of kernel methods comes from the fact that they implicitly map the data to a high dimensional, or even infinite dimensional, feature space, where points with different labels can be separated by a linear functional. It is, however, time-consuming to compute the kernel matrix and thus KSVMs do not scale well to extremely large datasets. To overcome this challenge, researchers have developed various ways to efficiently approximate the kernel matrix or the kernel function.

The random features method, proposed by Rahimi and Recht [2008], maps the data to a finite dimensional feature space as a random approximation to the feature space of RBF kernels. With explicit finite dimensional feature vectors available, the original KSVM is converted to a linear support vector machine (LSVM), that can be trained by faster algorithms (Shalev-Shwartz et al. [2011], Hsieh et al. [2008]) and tested in constant time with respect to the number of training samples. For example, Huang et al. [2014] and Dai et al. [2014] applied RFSVM or its variant to datasets containing millions of data points and achieved performance comparable to deep neural nets.

Despite solid practical performance, there is a lack of clear theoretical guarantees for the learning rate of RFSVM. Rahimi and Recht [2009] obtained a risk gap of order $O(1/\sqrt{N})$ between the best RFSVM and KSVM classifiers, where $N$ is the number of features. Although the order of the error bound is correct for general cases, it is too pessimistic to justify or to explain the actual computational benefits of random features method in practice. And the model is formulated as a constrained optimization problem, which is rarely used in practice.

Cortes et al. [2010] and Sutherland and Schneider [2015] considered the performance of RFSVM as a perturbed optimization problem, using the fact that the dual form of KSVM is a constrained quadratic optimization problem. Although the maximizer of a quadratic function depends continuously on the quadratic form, its dependence is weak and thus, both papers failed to obtain an informative bound for the excess risk of RFSVM in the classification problem. In particular, such an approach requires RFSVM and KSVM to be compared under the same hyper-parameters. This assumption is, in fact,

problematic because the optimal configuration of hyper-parameters of RFSVM is not necessarily the same as those for the corresponding KSVM. In this sense, RFSVM is more like an independent learning model instead of just an approximation to KSVM.

In regression settings, the learning rate of random features method was studied by Rudi and Rosasco [2017] under the assumption that the regression function is in the RKHS, namely the *realizable* case. They show that the uniform feature sampling only requires $O(\sqrt{m}\log(m))$ features to achieve $O(1/\sqrt{m})$ risk of squared loss. They further show that a data-dependent sampling can achieve a rate of $O(1/m^\alpha)$, where $1/2 \leq \alpha \leq 1$, with even fewer features, when the regression function is sufficiently smooth and the spectrum of the kernel integral operator decays sufficiently fast. However, the method leading to these results depends on the closed form of the least squares solution, and thus we cannot easily extend these results to non-smooth loss functions used in RFSVM. Bach [2017] recently shows that for any given approximation accuracy, the number of random features required is given by the degrees of freedom of the kernel operator under such an accuracy level, when optimized features are available. This result is crucial for sample complexity analysis of RFSVM, though not many details are provided on this topic in Bach's work.

In this paper, we investigate the performance of RFSVM formulated as a regularized optimization problem on classification tasks. In contrast to the slow learning rate in previous results by Rahimi and Recht [2009] and Bach [2017], we show, for the first time, that RFSVM can achieve fast learning rate with far fewer features than the number of samples when the optimized features (see Assumption 2) are available, and thus we justify the potential computational benefits of RFSVM on classification tasks. We mainly considered two learning scenarios: the realizable case, and then unrealizable case, where the Bayes classifier does not belong to the RKHS of the feature map. In particular, our contributions are threefold:

1. We prove that under Massart's low noise condition, with an optimized feature map, RFSVM can achieve a learning rate of $\tilde{O}(m^{-\frac{c_2}{1+c_2}})$ [1], with $\tilde{O}(m^{\frac{2}{2+c_2}})$ number of features when the Bayes classifier belongs to the RKHS of a kernel whose spectrum decays polynomially ($\lambda_i = O(i^{-c_2})$). When the decay rate of the spectrum of kernel operator is sub-exponential, the learning rate can be improved to $\tilde{O}(1/m)$ with only $\tilde{O}(\ln^d(m))$ number of features.

2. When the Bayes classifier satisfies the separation condition; that is, when the two classes of points are apart by a positive distance, we prove that the RFSVM using an optimized feature map corresponding to Gaussian kernel can achieve a learning rate of $\tilde{O}(1/m)$ with $\tilde{O}(\ln^{2d}(m))$ number of features.

3. Our theoretical analysis suggests reweighting random features before training. We confirm its benefit in our experiments over synthetic data sets.

We begin in Section 2 with a brief introduction of RKHS, random features and the problem formulation, and set up the notations we use throughout the rest of the paper. In Section 3, we provide our main theoretical results (see the appendices for the proofs), and in Section 4, we verify the performance of RFSVM in experiments. In particular, we show the improvement brought by the reweighted feature selection algorithm. The conclusion and some open questions are summarized at the end. The proofs of our main theorems follow from a combination of the sample complexity analysis scheme used by Steinwart and Christmann [2008] and the approximation error result of Bach [2017]. The fast rate is achieved due to the fact that the Rademacher complexity of the RKHS of $N$ random features and with regularization parameter $\lambda$ is only $O(\sqrt{N\log(1/\lambda)})$, while $N$ and $1/\lambda$ need not be too large to control the approximation error when optimized features are available. Detailed proofs and more experimental results are provided in the Appendices for interested readers.

## 2   Preliminaries and notations

Throughout this paper, a labeled data point is a point $(x, y)$ in $\mathcal{X} \times \{-1, 1\}$, where $\mathcal{X}$ is a bounded subset of $\mathbb{R}^d$. $\mathcal{X} \times \{-1, 1\}$ is equipped with a probability distribution $\mathbb{P}$.

## 2.1 Kernels and Random Features

A positive definite kernel function $k(x, x')$ defined on $\mathcal{X} \times \mathcal{X}$ determines the unique corresponding reproducing kernel Hilbert space (RKHS), denoted by $\mathcal{F}_k$. A map $\phi$ from the data space $\mathcal{X}$ to a Hilbert space $H$ such that $\langle \phi(x), \phi(x') \rangle_H = k(x, x')$ is called a feature map of $k$ and $H$ is called a feature space. For any $f \in \mathcal{F}$, there exists an $h \in H$ such that $\langle h, \phi(x) \rangle_H = f(x)$, and the infimum of the norms of all such $h$s is equal to $\|f\|_{\mathcal{F}}$. On the other hand, given any feature map $\phi$ into $H$, a kernel function is defined by the equation above, and we call $\mathcal{F}_k$ the RKHS corresponding to $\phi$, denoted by $\mathcal{F}_\phi$.

A common choice of feature space is the $L^2$ space of a probability space $(\omega, \Omega, \nu)$. An important observation is that for any probability density function $q(\omega)$ defined on $\Omega$, $\phi(\omega; x)/\sqrt{q(\omega)}$ with probability measure $q(\omega)\mathrm{d}\nu(\omega)$ defines the same kernel function with the feature map $\phi(\omega; x)$ under the distribution $\nu$. One can sample the image of $x$ under the feature map $\phi$, an $L^2$ function $\phi(\omega; x)$, at points $\{\omega_1, \ldots, \omega_N\}$ according to the probability distribution $\nu$ to approximately represent $x$. Then the vector in $\mathbb{R}^N$ is called a random feature vector of $x$, denoted by $\phi_N(x)$. The corresponding kernel function determined by $\phi_N$ is denoted by $k_N$.

A well-known construction of random features is the random Fourier features proposed by Rahimi and Recht [2008]. The feature map is defined as follows,

$$\phi : \mathcal{X} \to L^2(\mathbb{R}^d, \nu) \oplus L^2(\mathbb{R}^d, \nu)$$
$$x \mapsto (\cos(\omega \cdot x), \sin(\omega \cdot x)).$$

And the corresponding random feature vector is

$$\phi_N(x) = \frac{1}{\sqrt{N}} (\cos(\omega \cdot x), \cdots, \cos(\omega \cdot x), \sin(\omega \cdot x), \cdots, \sin(\omega \cdot x))^\mathsf{T},$$

where $\omega_i$s are sampled according to $\nu$. Different choices of $\nu$ define different translation invariant kernels (see Rahimi and Recht [2008]). When $\nu$ is the normal distribution with mean 0 and variance $\gamma^{-2}$, the kernel function defined by the feature map is Gaussian kernel with bandwidth parameter $\gamma$,

$$k_\gamma(x, x') = \exp\left(-\frac{\|x - x'\|^2}{2\gamma^2}\right).$$

Equivalently, we may consider the feature map $\phi_\gamma(\omega; x) := \phi(\omega/\gamma; x)$ with $\nu$ being standard normal distribution.

A more general and more abstract feature map can be constructed using an orthonormal set of $L^2(\mathcal{X}, \mathbb{P}_\mathcal{X})$. Given the orthonormal set $\{e_i\}$ consisting of bounded functions, and a nonnegative sequence $(\lambda_i) \in \ell^1$, we can define a feature map

$$\phi(\omega; x) = \sum_{i=1}^{\infty} \sqrt{\lambda_i} e_i(x) e_i(\omega),$$

with feature space $L^2(\omega, \mathcal{X}, \mathbb{P}_\mathcal{X})$. The corresponding kernel is given by $k(x, x') = \sum_{i=1}^{\infty} \lambda_i e_i(x) e_i(x')$. The feature map and the kernel function are well defined because of the boundedness assumption on $\{e_i\}$. A similar representation can be obtained for a continuous kernel function on a compact set by Mercer's Theorem (Lax [2002]).

Every positive definite kernel function $k$ satisfying that $\int k(x, x) \, \mathrm{d}\mathbb{P}_\mathcal{X}(x) < \infty$ defines an integral operator on $L^2(x, \mathcal{X}, \mathbb{P}_\mathcal{X})$ by

$$\Sigma : L^2(\mathcal{X}, \mathbb{P}_\mathcal{X}) \to L^2(\mathcal{X}, \mathbb{P}_\mathcal{X})$$
$$f \mapsto \int_\mathcal{X} k(x, t) f(t) \, \mathrm{d}\mathbb{P}_\mathcal{X}(t).$$

$\Sigma$ is of trace class with trace norm $\int k(x, x) \, \mathrm{d}\mathbb{P}_\mathcal{X}(x)$. When the integral operator is determined by a feature map $\phi$, we denote it by $\Sigma_\phi$, and the $i$th eigenvalue in a descending order by $\lambda_i(\Sigma_\phi)$. Note that the regularization paramter is also denoted by $\lambda$ but without a subscript. The decay rate of the spectrum of $\Sigma_\phi$ plays an important role in the analysis of learning rate of random features method.

## 2.2 Formulation of Support Vector Machine

Given $m$ samples $\{(x_i, y_i)\}_{i=1}^m$ generated i.i.d. by $\mathbb{P}$ and a function $f : \mathcal{X} \to \mathbb{R}$, usually called a hypothesis in the machine learning context, the empirical and expected risks with respect to the loss function $\ell$ are defined by

$$R_m^\ell(f) := \frac{1}{m} \sum_{i=1}^m \ell(y_i, f(x_i)) \quad R_{\mathbb{P}}^\ell(f) := \mathbb{E}_{(x,y) \sim \mathbb{P}} \ell(y, f(x)) ,$$

respectively.

The 0-1 loss is commonly used to measure the performance of classifiers:

$$\ell^{0-1}(y, f(x)) = \begin{cases} 1 & \text{if } f(x)y \le 0; \\ 0 & \text{if } f(x)y > 0. \end{cases}$$

The function that minimizes the expected risk under 0-1 loss is called the Bayes classifier, defined by

$$f_{\mathbb{P}}^*(x) := \operatorname{sgn}(\mathbb{E}[y \mid x]) .$$

The goal of the classification task is to find a good hypothesis $f$ with small excess risk $R_{\mathbb{P}}^{0-1}(f) - R_{\mathbb{P}}^{0-1}(f_{\mathbb{P}}^*)$. And to find the good hypothesis based on the samples, one minimizes the empirical risk. However, using 0-1 loss, it is hard to find the global minimizer of the empirical risk because the loss function is discontinuous and non-convex. A popular surrogate loss function in practice is the hinge loss: $\ell^h(f) = \max(0, 1 - yf(x))$, which guarantees that

$$R_{\mathbb{P}}^h(f) - \inf_f R_{\mathbb{P}}^h(f) \ge R_{\mathbb{P}}^{0-1}(f) - R_{\mathbb{P}}^{0-1}(f_{\mathbb{P}}^*),$$

where $R^h$ means $R^{\ell^h}$ and $R^{0-1}$ means $R^{\ell^{0-1}}$. See Steinwart and Christmann [2008] for more details.

A regularizer can be added into the optimization objective with a scalar multiplier $\lambda$ to avoid overfitting the random samples. Throughout this paper, we consider the most commonly used $\ell^2$ regularization. Therefore, the solution of the binary classification problem is given by minimizing the following objective

$$R_{m,\lambda}(f) = R_m^h(f) + \frac{\lambda}{2} \|f\|_{\mathcal{F}}^2 ,$$

over a hypothesis class $\mathcal{F}$. When $\mathcal{F}$ is the RKHS of some kernel function, the algorithm described above is called kernel support vector machine. Note that for technical convenience, we do not include the bias term in the formulation of hypothesis so that all these functions are from the RKHS instead of the product space of RKHS and $\mathbb{R}$ (see Chapter 1 of Steinwart and Christmann [2008] for more explanation of such a convention). Note that $R_{m,\lambda}$ is strongly convex and thus the infimum will be attained by some function in $\mathcal{F}$. We denote it by $f_{m,\lambda}$.

When random features $\phi_N$ and the corresponding RKHS are considered, we add $N$ into the subscripts of the notations defined above to indicate the number of random features. For example $\mathcal{F}_N$ for the RKHS, $f_{N,m,\lambda}$ for the solution of the optimization problem.

## 3 Main Results

In this section we state our main results on the fast learning rates of RFSVM in different scenarios.

First, we need the following assumption on the distribution of data, which is required for all the results in this paper.

**Assumption 1.** *There exists $V \ge 2$ such that*

$$|\mathbb{E}_{(x,y) \sim \mathbb{P}}[y \mid x]| \ge 2/V .$$

This assumption is called Massart's low noise condition in many references (see for example Koltchinskii et al. [2011]). When $V = 2$ then all the data points have deterministic labels almost surely. Therefore it is easier to learn the true classifier based on observations. In the proof, Massart's low noise condition guarantees the variance condition (Steinwart and Christmann [2008])

$$\mathbb{E}[(\ell^h(f(x)) - \ell^h(f_{\mathbb{P}}^*(x)))^2] \le V(R^h(f) - R^h(f_{\mathbb{P}}^*)), \tag{1}$$

which is a common requirement for the fast rate results. Massart's condition is an extreme case of a more general low noise condition, called Tsybakov's condition. For the simplicity of the theorem, we only consider Massart's condition in our work, but our main results can be generalized to Tsybakov's condition.

The second assumption is about the quality of random features. It was first introduced in Bach [2017]'s approximation results.

**Assumption 2.** *A feature map $\phi : \mathcal{X} \to L^2(\omega, \Omega, \nu))$ is called optimized if there exists a small constant $\mu_0$ such that for any $\mu \leq \mu_0$,*

$$\sup_{\omega \in \Omega} \|(\Sigma + \mu I)^{-1/2} \phi(\omega; x)\|_{L^2(\mathbb{P})}^2 \leq \mathrm{tr}(\Sigma(\Sigma + \mu I)^{-1}) = \sum_{i=1}^{\infty} \frac{\lambda_i(\Sigma)}{\lambda_i(\Sigma) + \mu} \,.$$

For any given $\mu$, the quantity on the left hand side of the inequality is called leverage score with respect to $\mu$, which is directly related with the number of features required to approximate a function in the RKHS of $\phi$. The quantity on the right hand side is called degrees of freedom by Bach [2017] and effective dimension by Rudi and Rosasco [2017], denoted by $d(\mu)$. Note that whatever the RKHS is, we can always construct optimized feature map for it. In the Appendix A we describe two examples of constructing optimized feature map. When a feature map is optimized, it is easy to control its leverage score by the decay rate of the spectrum of $\Sigma$, as described below.

**Definition 1.** *We say that the spectrum of $\Sigma : L^2(\mathcal{X}, \mathbb{P}) \to L^2(\mathcal{X}, \mathbb{P})$ decays at a polynomial rate if there exist $c_1 > 0$ and $c_2 > 1$ such that*

$$\lambda_i(\Sigma) \leq c_1 i^{-c_2} \,.$$

*We say that it decays sub-exponentially if there exist $c_3, c_4 > 0$ such that*

$$\lambda_i(\Sigma) \leq c_3 \exp(-c_4 i^{1/d}) \,.$$

The decay rate of the spectrum of $\Sigma$ characterizes the capacity of the hypothesis space to search for the solution, which further determines the number of random features required in the learning process. Indeed, when the feature map is optimized, the number of features required to approximate a function in the RKHS with accuracy $O(\sqrt{\mu})$ is upper bounded by $O(d(\mu) \ln(d(\mu)))$. When the spectrum decays polynomially, the degrees of freedom $d(\mu)$ is $O(\mu^{-1/c_2})$, and when it decays sub-exponentially, $d(\mu)$ is $O(\ln^d(c_3/\mu))$ (see Lemma 6 in Appendix C for details). Examples on the kernels with polynomial and sub-exponential spectrum decays can be found in Bach [2017]. Our proof of Lemma 8 also provides some useful discussion.

With these preparations, we can state our first theorem now.

**Theorem 1.** *Assume that $\mathbb{P}$ satisfies Assumption 1, and the feature map $\phi$ satisfies Assumption 2. If $f_{\mathbb{P}}^* \in \mathcal{F}_\phi$ with $\|f_{\mathbb{P}}^*\|_{\mathcal{F}_\phi} \leq R$. Then when the spectrum of $\Sigma_\phi$ decays polynomially, by choosing*

$$\lambda = m^{-\frac{c_2}{2+c_2}}$$
$$N = 10 C_{c_1, c_2} m^{\frac{2}{2+c_2}} (\ln(32 C_{c_1, c_2} m^{\frac{2}{2+c_2}}) + \ln(1/\delta)) \,,$$

*we have*

$$R_{\mathbb{P}}^{0-1}(f_{N,m,\lambda}) - R_{\mathbb{P}}^{0-1}(f_{\mathbb{P}}^*) \leq C_{c_1, c_2, V, R} m^{-\frac{c_2}{2+c_2}} ((\ln(1/\delta) + \ln(m))) \,,$$

*with probability $1 - 4\delta$. When the spectrum of $\Sigma_\phi$ decays sub-exponentially, by choosing*

$$\lambda = 1/m$$
$$N = 25 C_{d, c_4} \ln^d(m)(\ln(80 C_{d, c_4} \ln^d(m)) + \ln(1/\delta)) \,,$$

*we have*

$$R_{\mathbb{P}}^{0-1}(f_{N,m,\lambda}) - R_{\mathbb{P}}^{0-1}(f_{\mathbb{P}}^*) \leq C_{c_3, c_4, d, R, V} \frac{1}{m} \left( \log^{d+2}(m) + \log(1/\delta) \right) \,,$$

*with probability $1 - 4\delta$ when $m \geq \exp((c_4 \vee \frac{1}{c_4}) d^2 / 2)$.*

This theorem characterizes the learning rate of RFSVM in realizable cases; that is, when the Bayes classifier belongs to the RKHS of the feature map. For polynomially decaying spectrum, when $c_2 > 2$, we get a learning rate faster than $1/\sqrt{m}$. Rudi and Rosasco [2017] obtained a similar fast learning rate for kernel ridge regression with random features (RFKRR), assuming polynomial decay of the spectrum of $\Sigma_\phi$ and the existence of a minimizer of the risk in $\mathcal{F}_\phi$. Our theorem extends their result to classification problems and exponential decay spectrum. However, we have to use a stronger assumption that $f_{\mathbb{P}}^* \in \mathcal{F}_\phi$ so that the low noise condition can be applied to derive the variance condition. For RFKRR, the rate faster than $O(1/\sqrt{m})$ will be achieved whenever $c_2 > 1$, and the number of features required is only square root of our result. We think that this is mainly caused by the fact that their surrogate loss is squared. The result for the sub-exponentially decaying spectrum is not investigated for RFKRR, so we cannot make a comparison. We believe that this is the first result showing that RFSVM can achieve $\tilde{O}(1/m)$ with only $\tilde{O}(\ln^d(m))$ features. Note however that when $d$ is large, the sub-exponential case requires a large number of samples, even possibly larger than the polynomial case. This is clearly an artifact of our analysis since we can always use the polynomial case to provide an upper bound! We therefore suspect that there is considerable room for improving our analysis of high dimensional data in the sub-exponential decay case. In particular, removing the exponential dependence on $d$ under reasonable assumptions is an interesting direction for future work.

To remove the realizability assumption, we provide our second theorem, on the learning rate of RFSVM in unrealizable case. We focus on the random features corresponding to the Gaussian kernel as introduced in Section 2. When the Bayes classifier does not belong to the RKHS, we need an approximation theorem to estimate the gap of risks. The approximation property of RKHS of Gaussian kernel has been studied in Steinwart and Christmann [2008], where the margin noise exponent is defined to derive the risk gap. Here we introduce the simpler and stronger separation condition, which leads to a strong result.

The points in $\mathcal{X}$ can be collected in to two sets according to their labels as follows,

$$\mathcal{X}_1 := \{x \in \mathcal{X} \mid \mathbb{E}(y \mid x) > 0\}$$
$$\mathcal{X}_{-1} := \{x \in \mathcal{X} \mid \mathbb{E}(y \mid x) < 0\}.$$

The distance of a point $x \in \mathcal{X}_i$ to the set $\mathcal{X}_{-i}$ is denoted by $\Delta(x)$.

**Assumption 3.** *We say that the data distribution satisfies a separation condition if there exists $\tau > 0$ such that $\mathbb{P}_{\mathcal{X}}(\Delta(x) < \tau) = 0$.*

Intuitively, Assumption 3 requires the two classes to be far apart from each other almost surely. This separation assumption is an extreme case when the margin noise exponent goes to infinity.

The separation condition characterizes a different aspect of data distribution from Massart's low noise condition. Massart's low noise condition guarantees that the random samples represent the distribution behind them accurately, while the separation condition guarantees the existence of a smooth, in the sense of small derivatives, function achieving the same risk with the Bayes classifier.

With both assumptions imposed on $\mathbb{P}$, we can get a fast learning rate of $\ln^{2d+1} m/m$ with only $\ln^{2d}(m)$ random features, as stated in the following theorem.

**Theorem 2.** *Assume that $\mathcal{X}$ is bounded by radius $\rho$. The data distribution has density function upper bounded by a constant $B$, and satisfies Assumption 1 and 3. Then by choosing*

$$\lambda = 1/m \quad \gamma = \tau/\sqrt{\ln m} \quad N = C_{\tau,d,\rho} \ln^{2d} m(\ln\ln m + \ln(1/\delta)),$$

*the RFSVM using an optimized feature map corresponding to the Gaussian kernel with bandwidth $\gamma$ achieves the learning rate*

$$R_{\mathbb{P}}^{0-1}(f_{N,m,\lambda}) - R_{\mathbb{P}}^{0-1}(f_{\mathbb{P}}^*) \leq C_{\tau,V,d,\rho,B} \frac{\ln^{2d+1}(m)(\ln\ln(m) + \ln(1/\delta))}{m},$$

*with probability greater than $1 - 4\delta$ for $m \geq m_0$, where $m_0$ depends on $\tau, \rho, d$.*

To the best of our knowledge, this is the first theorem on the fast learning rate of random features method in the unrealizable case. It only assumes that the data distribution satisfies low noise and separation conditions, and shows that with an optimized feature distribution, the learning rate of

$\tilde{O}(1/m)$ can be achieved using only $\ln^{2d+1}(m) \ll m$ features. This justifies the benefit of using RFSVM in binary classification problems. The assumption of a bounded data set and a bounded distribution density function can be dropped if we assume that the probability density function is upper bounded by $C \exp(-\gamma^2 \|x\|^2/2)$, which suffices to provide the sub-exponential decay of spectrum of $\Sigma_\phi$. But we prefer the simpler form of the results under current conditions. We speculate that the conclusion of Theorem 2 can be generalized to all sub-Gaussian data.

The main drawback of our two theorems is the assumption of an optimized feature distribution, which is hard to obtain in practice. Developing a data-dependent feature selection method is therefore an important problem for future work on RFSVM. Bach [2017] proposed an algorithm to approximate the optimized feature map from any feature map. Adapted to our setup, the reweighted feature selection algorithm is described as follows.

1. Select $M$ i.i.d. random vectors $\{\omega_i\}_{i=1}^M$ according to the distribution $d\nu_\gamma$.

2. Select $L$ data points $\{x_i\}_{i=1}^L$ uniformly from the training set.

3. Generate the matrix $\Phi$ with columns $\phi_M(x_i)/\sqrt{L}$.

4. Compute $\{r_i\}_{i=1}^M$, the diagonal of $\Phi\Phi^\intercal(\Phi\Phi^\intercal + \mu I)^{-1}$.

5. Resample $N$ features from $\{\omega_i\}_{i=1}^M$ according to the probability distribution $p_i = r_i/\sum r_i$.

The theoretical guarantees of this algorithm have not been discussed in the literature. A result in this direction will be extremely useful for guiding practioners. However, it is outside the scope of our work. Instead, here we implement it in our experiment and empirically compare the performance of RFSVM using this reweighted feature selection method to the performance of RFSVM without this preprocessing step; see Section 4.

For the realizable case, if we drop the assumption of optimized feature map, only weak results can be obtained for the learning rate and the number of features required (see Appendix E for more details). In particular, we can only show that $1/\epsilon^2$ random features are sufficient to guarantee the learning rate less than $\epsilon$ when $1/\epsilon^3$ samples are available. Though not helpful for justifying the computational benefit of random features method, this result matches the parallel result for RFKRR in Rudi and Rosasco [2017] and the approximation result in Sriperumbudur and Szabo [2015]. We conjecture that this upper bound is also optimal for RFSVM.

Rudi and Rosasco [2017] also compared the performance of RFKRR with Nystrom method, which is the other popular method to scale kernel ridge regression to large data sets. We do not find any theoretical guarantees on the fast learning rate of SVM with Nystrom method on classification problems in the literature, though there are several works on its approximation quality to the accurate model and its empirical performance (see Yang et al. [2012], Zhang et al. [2012]). The tools used in this paper should also work for learning rate analysis of SVM using Nystrom method. We leave this analysis to the future.

## 4 Experimental Results

In this section we evaluate the performance of RFSVM with the reweighted feature selection algorithm[2]. The sample points shown in Figure 3 are generated from either the inner circle or outer annulus uniformly with equal probability, where the radius of the inner circle is 0.9, and the radius of the outer annulus ranges from 1.1 to 2. The points from the inner circle are labeled by -1 with probability 0.9, while the points from the outer annulus are labeled by 1 with probability 0.9. In such a simple case, the unit circle describes the Bayes classifier.

First, we compared the performance of RFSVM with that of KSVM on the training set with 1000 samples, over a large range of regularization parameter ($-7 \leq \log\lambda \leq 1$). The bandwidth parameter $\gamma$ is fixed to be an estimate of the average distance among the training samples. After training, models are tested on a large testing set ($> 10^5$). For RFSVM, we considered the effect of the number of features by setting $N$ to be $1, 3, 5, 10$ and $20$, respectively. Moreover, both feature selection methods, simple random feature selection (labeled by 'unif' in the figures), which does not apply any preprocess on drawing features, and reweighted feature selection (labeled by 'opt' in the figures) are

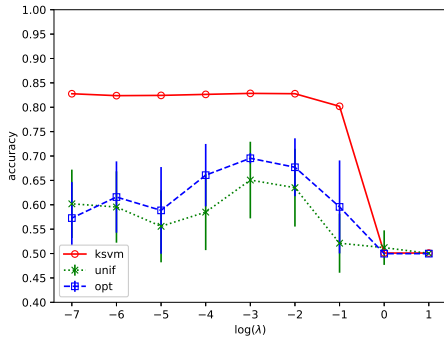

Figure 1: RFSVM with 1 feature.

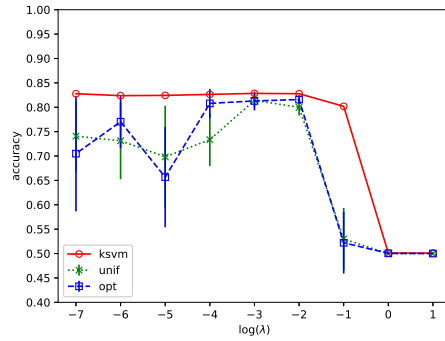

Figure 2: RFSVM with 20 features.

"ksvm" is for KSVM with Gaussian kernel, "unif" is for RFSVM with direct feature sampling, and "opt" is for RFSVM with reweighted feature sampling. Error bars represent standard deviation over 10 runs.

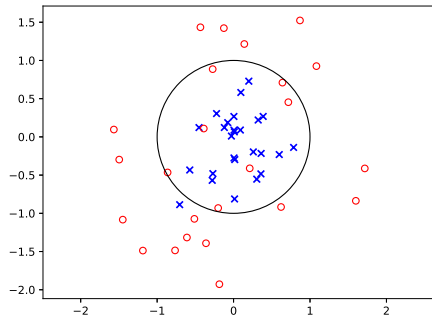

Figure 3: Distribution of Training Samples.

50 points are shown in the graph. Blue crosses represent the points labeled by -1, and red circles the points labeled by 1. The unit circle is one of the best classifier for these data with 90% accuracy.

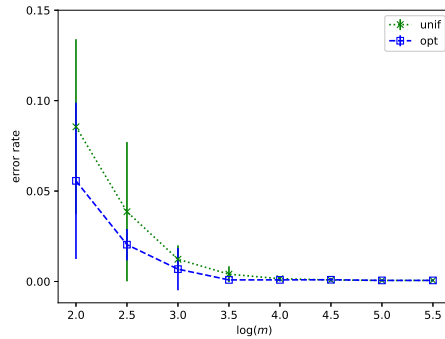

Figure 4: Learning Rate of RFSVMs.

The excess risks of RFSVMs with the simple random feature selection ("unif") and the reweighted feature selection ("opt") are shown for different sample sizes. The error rate is the excess risk. The error bars represent the standard deviation over 10 runs.

inspected. For the reweighted method, we set $M = 100N$ and $L = 0.3m$ to compute the weight of each feature. Every RFSVM is run 10 times, and the average accuracy and standard deviation are presented.

The results of KSVM, RFSVMs with 1 and 20 features are shown in Figure 1 and Figure 2 respectively (see the results of other levels of features in Appendix F in the supplementary material). The performance of RFSVM is slightly worse than the KSVM, but improves as the number of features increases. It also performs better when the reweighted method is applied to generate features.

To further compare the performance of simple feature selection and reweighted feature selection methods, we plot the learning rate of RFSVM with $O(\ln^2(m))$ features and the best $\lambda$s for each sample size $m$. KSVM is not included here since it is too slow on training sets of size larger than $10^4$ in our experiment compared to RFSVM. The error rate in Figure 4 is the excess risk between learned classifiers and the Bayes classifier. We can see that the excess risk decays as $m$ increases, and the RFSVM using reweighted feature selection method outperforms the simple feature selection.

According to Theorem 2, the benefit brought by optimized feature map, that is, the fast learning rate, will show up when the sample size is greater than $O(\exp(d))$ (see Appendix D). The number of random features required also depends on $d$, the dimension of data. For data of small dimension and large sample size, as in our experiment, it is not a problem. However, in applications of image

recognition, the dimension of the data is usually very large and it is hard for our theorem to explain the performance of RFSVM. On the other hand, if we do not pursue the fast learning rate, the analysis for general feature maps, not necessarily optimized, gives a learning rate of $O(m^{-1/3})$ with $O(m^{2/3})$ random features, which does not depend on the dimension of data (see Appendix E). Actually, for high dimensional data, there is barely any improvement in the performance of RFSVM by using reweighted feature selection method (see Appendix F). It is important to understand the role of $d$ to fully understand the power of random features method.

## 5 Conclusion

Our study proves that the fast learning rate is possible for RFSVM in both realizable and unrealizable scenarios when the optimized feature map is available. In particular, the number of features required is far less than the sample size, which implies considerably faster training and testing using the random features method. Moreover, we show in the experiments that even though we can only approximate the optimized feature distribution using the reweighted feature selection method, it, indeed, has better performance than the simple random feature selection. Considering that such a reweighted method does not rely on the label distribution at all, it will be useful in learning scenarios where multiple classification problems share the same features but differ in the class labels. We believe that a theoretical guarantee of the performance of the reweighted feature selection method and properly understanding the dependence on the dimensionality of data are interesting directions for future work.

**Acknowledgements**

AT acknowledges the support of a Sloan Research Fellowship.

ACG acknowledges the support of a Simons Foundation Fellowship.

## Footnotes

[1] $\tilde{O}(n)$ represents a quantity less than $Cn\log^k(n)$ for some $k$.

[2]The source code is available at `https://github.com/syitong/randfourier`.

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
