[Supplementary Material]

## A    Examples of Optimized Feature Maps

Assume that a feature map $\phi : (X) \to L^2(\omega, \Omega, \nu)$ satisfies that $\phi(\omega; x)$ is bounded for all $\omega$ and $x$. We can always convert it to an optimized feature map using the method proposed by Bach [2017]. We rephrase it using our notation as follows.

Define

$$p(\omega) = \frac{\|(\Sigma + \mu I)^{-1/2}\phi(\cdot; \omega)\|^2_{L^2(\mathcal{X}, \mathbb{P})}}{\int_\Omega \|(\Sigma + \mu I)^{-1/2}\phi(\cdot; \omega)\|^2_{L^2(\mathcal{X}, \mathbb{P})}\, \mathrm{d}\nu(\omega)}\,. \tag{2}$$

Since $\phi$ is bounded, its $L^2$ norm is finite. The function $p$ defined above is a probability density function with respect to $\nu$. Then the new feature map is given by $\tilde{\phi}(\omega; x) = \phi(\omega; x)/\sqrt{p(\omega)}$ together with the measure $p(\omega)\mathrm{d}\nu(\omega)$. With $\tilde{\phi}$, we have

$$\sup_{\omega \in \Omega} \left\|(\Sigma + \mu I)^{-1/2}\tilde{\phi}(\cdot; \omega)\right\|^2 = \sup_{\omega \in \Omega} \frac{\left\|(\Sigma + \mu I)^{-1/2}\phi(\cdot; \omega)\right\|^2}{p(\omega)} \tag{3}$$

$$= \int_\Omega \|(\Sigma + \mu I)^{-1/2}\phi(\cdot; \omega)\|^2_{L^2(\mathcal{X}, \mathbb{P})}\, \mathrm{d}\nu(\omega) \tag{4}$$

$$= \operatorname{tr}(\Sigma(\Sigma + \mu I)^{-1})\,. \tag{5}$$

When the feature map is constructed mapping into $L^2(\mathcal{X}, \mathbb{P})$ as described in Section 2, it is optimized. Indeed, we can compute

$$\sup_{\omega \in \mathcal{X}} \left\|(\Sigma + \mu I)^{-1/2}\phi(\cdot; \omega)\right\|^2 = \sup_{\omega \in \mathcal{X}} \left\|\sum_{i=1}^\infty \frac{\sqrt{\lambda_i}}{\sqrt{\lambda_i + \mu}} e_i(\cdot)\right\|^2 \tag{6}$$

$$= \sum_{i=1}^\infty \frac{\lambda_i}{\lambda_i + \mu}\,. \tag{7}$$

As an example for this type of feature map, we can consider $\{e_i\}$ to be the Walsh system, which is an orthonormal basis for $L^2([0, 1])$. Any Bayes classifier with finitely many discontinuities and discontinuous only at dyadic, namely points expressable by finite bits, points, will be a finite linear combination of Walsh basis. This guarantees that the assumptions in Theorem 1 can be satisfied. Our first experiment also make use of this construction.

The construction above is inspired by the use of spline kernel in Rudi and Rosasco [2017]. However, our situation is more complicated since the target function, Bayes classifier, is discontinuous. While the functions in the RKHS generated by the spline kernel must be continuous (Cucker and Smale [2002]). Though we can construct Bayes classifier using the Walsh basis, we have yet to understand the variety of possible Bayes classifiers in such a space.

## B    Local Rademacher Complexity of RFSVM

Before the proofs, we first briefly summarize the use of each lemmas and theorems. Theorem 3 and 4 are two fundamental external results for our proof. Lemma 7 and 8 refine results that appeared in previous works, so that we can apply them to our case. Lemma 3, 4 and 5 are the key results to establish fast rate for RFSVM, parallel to Steinwarts' work for KSVM. All other smaller and simpler lemmas included in the appendices are for the purposes of clarity and completeness. The proofs are not hard but quite technical.

First, both of our theorems are consequences of the following fundamental theorem. In the theorem, $\ell^1$ is the hinge loss clipped at 1, and $R^1_{\mathbb{P}, \lambda}$ is the expected regularized risk of $\ell^1$.

**Theorem 3.** *(Theorem 7.20 in Steinwart and Christmann [2008]) For a RKHS $\mathcal{F}$, denote $\inf_{f \in \mathcal{F}} R^1_{\mathbb{P}, \lambda}(f) - R^*$ by $r^*$. For $r > r^*$, consider the following function classes*

$$\mathcal{F}_r := \{f \in \mathcal{F} \mid R^1_{\mathbb{P}, \lambda}(f) - R^* \leq r\}$$

*and*

$$\mathcal{H}_r := \{\ell^1 \circ f - \ell^1 \circ f^*_{\mathbb{P}} \mid f \in \mathcal{F}_r\}\,.$$

*Assume that there exists $V \geq 1$ such that for any $f \in \mathcal{F}$,*

$$\mathbb{E}_{\mathbb{P}}(\ell^1 \circ f - \ell^1 \circ f_{\mathbb{P}}^*)^2 \leq V(R_{\mathbb{P}}^1(f) - R^*).$$

*If there is a function $\varphi_m : [0, \infty) \to [0, \infty)$ such that $\varphi_m(4r) \leq 2\varphi_m(r)$ and $\mathfrak{R}_m(\mathcal{H}_r) \leq \varphi_m(r)$ for all $r \geq r^*$, Then, for any $\delta \in (0, 1]$, $f_0 \in \mathcal{F}$ with $\|\ell^{\text{hinge}} \circ f_0\|_\infty \leq B_0$, and*

$$r > \max\left\{30\varphi_m(r), \frac{72V \ln(1/\delta)}{m}, \frac{5B_0 \ln(1/\delta)}{m}, r^*\right\},$$

*we have*

$$R_{\mathbb{P},\lambda}^1(f_{m,N,\lambda}) - R^* \leq 6\left(R_{\mathbb{P},\lambda}^h(f_0) - R^*\right) + 3r$$

*with probability greater than $1 - 3\delta$.*

To establish the fast rate of RFSVM using the theorem above, we must understand the local Rademacher complexity of RFSVM: that is, find a formula for $\varphi_m(r)$. $B_0, r^*$ and $f_0$ are only related with the approximation error, and we leave the discussion of them to next sections. The variance condition Equation 1 is satisfied under Assumption 1. With this variance condition, we can upper bound the Rademacher complexity of RFSVM in terms of number of features and regularization parameter. It is particularly important to have $1/\lambda$ inside the logarithm function.

First, we will need the summation version of Dudley's inequality using entropy number defined below, instead of covering number.

**Definition 2.** *For a semi-normed space $(E, \|\cdot\|)$, we define its (dyadic) entropy number by*

$$e_n(E, \|\cdot\|) := \inf\left\{\varepsilon > 0 : \exists s_1, \ldots, s_{2^{n-1}} \in B^1 \text{ s.t. } B^1 \subset \bigcup_{i=1}^{2^{n-1}} B(s_i, \varepsilon)\right\},$$

*where $B^1$ is the unit ball in $E$ and $B(a, r)$ is the ball with center at $a$ and radius $r$.*

To take off the loss function from the hypothesis class, we have the following lemma. $\|\cdot\|_{L_2(D)}$ is the semi-norm defined by $\|\cdot\|_{L_2(D)} := (\frac{1}{m}\sum_i f^2(x_i))^{1/2}$.

**Lemma 1.** $e_i(\mathcal{H}_r, \|\cdot\|_{L_2(D)}) \leq e_i(\mathcal{F}_r, \|\cdot\|_{L_2(D)})$

*Proof.* Assume that $T$ is an $\epsilon$-covering over $\mathcal{F}_r$ with $|T| = 2^i$. By definition $\epsilon \geq e_i(\mathcal{F}_r, \|\cdot\|_{L_2(D)})$. Then $T' = \ell^1 \circ T - \ell^1 \circ f_{\mathbb{P}}^*$ is a covering over $\mathcal{H}_r$. For any $f$ and $g$ in $\mathcal{F}_r$,

$$\left\|\ell^1 \circ f - \ell^1 \circ g\right\|_{L_2(D)} \leq 1 \cdot \|f - g\|_{L_2(D)},$$

because $\ell^1$ is 1-Lipschitz. And hence the radius of the image of an $\epsilon$-ball under $\ell^1$ is less than $\epsilon$. Therefore $\ell^1 \circ T - \ell^1 \circ f_{\mathbb{P}}^*$ is an $\epsilon$-covering over $\mathcal{H}_r$ with cardinatily $2^i$ and $\epsilon \leq e_i(\mathcal{F}_r, \|\cdot\|_{L_2(D)})$. By taking infimum over the radius of all such $T$ and $T'$, the statement is proved. $\square$

Now we need to give an upper bound for the entropy number of $\mathcal{F}_r$ with semi-norm $\|\cdot\|_{L_2(D)}$ using a volumetric estimate.

**Lemma 2.** $e_i(\mathcal{F}_r, \|\cdot\|_{L_2(D)}) \leq 3(2r/\lambda)^{1/2}2^{-i/2N}$.

*Proof.* Since $\mathcal{F}$ consists of functions

$$f(x) = \frac{1}{\sqrt{N}}\sum_{i=1}^N w_{c_i}\cos\left(\frac{\omega_i \cdot x}{\gamma}\right) + w_{s_i}\sin\left(\frac{\omega_i \cdot x}{\gamma}\right),$$

under the semi-norm $\|\cdot\|_{L_2(D)}$ it is isometric with the $2N$-dimensional subspace $U$ of $\mathbb{R}^m$ spanned by the vectors

$$\left\{\left[\cos\left(\frac{\omega_i \cdot x_1}{\gamma}\right), \ldots, \cos\left(\frac{\omega_i \cdot x_m}{\gamma}\right)\right]^\intercal, \left[\sin\left(\frac{\omega_i \cdot x_1}{\gamma}\right), \ldots, \sin\left(\frac{\omega_i \cdot x_m}{\gamma}\right)\right]^\intercal\right\}_{i=1}^N$$

for fixed $m$ samples. For each $f \in \mathcal{F}_r$, we have $R_{\mathbb{P},\lambda}^1(f) - R^* \le r$, which implies that $\|f\|_{\mathcal{F}} \le (2r/\lambda)^{1/2}$. By the property of RKHS, we get

$$|f(x)| \le \|f\|_{\mathcal{F}} \|k(x,\cdot)\|_{\mathcal{F}} \le \left( \frac{2r}{\lambda} \right)^{1/2} \cdot 1 ,$$

where we use the fact that $k(x,\cdot)$ is the evaluation functional in the RKHS.

Denote the isomorphism from $\mathcal{F}$ (modulo the equivalent class under the semi-norm) to $U$ by $I$. Then we have

$$I(\mathcal{F}_r) \subset B_\infty^m \left( \left( \frac{2r}{m\lambda} \right)^{1/2} \right) \cap U \subset B_2^m \left( \left( \frac{2r}{\lambda} \right)^{1/2} \right) \cap U .$$

The intersection region can be identified as a ball of radius $(2r/\lambda)^{1/2}$ in $\mathbb{R}^{2N}$. Its entropy number by volumetric estimate is given by

$$e_i \left( B_2^{2N} \left( \left( \frac{2r}{\lambda} \right)^{1/2} \right), \|\cdot\|_2 \right) \le 3 \left( \frac{2r}{\lambda} \right)^{1/2} 2^{-\frac{i}{2N}} .$$

$\square$

With the lemmas above, we can get an upper bound on the entropy number of $\mathcal{H}_r$. However, we should note that such an upper bound is not the best when $i$ is small. Because the ramp loss $\ell^1$ is bounded by 2, the radius of $\mathcal{H}_r$ with respect to $\|\cdot\|_{L_2(D)}$ is bounded by 1, which is irrelevant with $r/\lambda$. This observation will give us finer control on the Rademacher complexity.

**Lemma 3.** *Assume that $\lambda < 1/2$. Then*

$$\mathfrak{R}_D(\mathcal{H}_r) \le \sqrt{\frac{(\ln 16) N \log_2 1/\lambda}{m}} \left( 3\sqrt{2}\rho + 18\sqrt{r} \right) ,$$

*where $\rho = \sup_{h \in \mathcal{H}_r} \|h\|_{L_2(D)}$.*

*Proof.* By Theorem 7.13 in Steinwart and Christmann [2008], we have

$$\mathfrak{R}_D(\mathcal{H}_r) \le \sqrt{\frac{\ln 16}{m}} \left( \sum_{i=1}^{\infty} 2^{i/2} e_{2^i}(\mathcal{H}_r \cup \{0\}, \|\cdot\|_{L_2(D)}) + \sup_{h \in \mathcal{H}_r} \|h\|_{L_2(D)} \right) .$$

It is easy to see that $e_i(\mathcal{H}_r \cup \{0\}) \le e_{i-1}(\mathcal{H}_r)$ and $e_0(\mathcal{H}_r) \le \sup_{h \in \mathcal{H}_r} \|h\|_{L_2(D)}$. Since $e_i(\mathcal{H}_r)$ is a decreasing sequence with respect to $i$, together with the lemma above, we know that

$$e_i(\mathcal{H}_r) \le \min \left\{ \sup_{h \in \mathcal{H}_r} \|h\|_{L_2(D)}, 3 \left( \frac{2r}{\lambda} \right)^{1/2} 2^{-\frac{i}{2N}} \right\} .$$

Even though the second one decays exponentially, it may be much greater than the first term when $2r/\lambda$ is huge for small $i$s. To achieve the balance between these two bounds, we use the first one for first $T$ terms in the sum and the second one for the tail. So

$$\mathfrak{R}_D(\mathcal{H}_r) \le \sqrt{\frac{\ln 16}{m}} \left( \sup_{h \in \mathcal{H}_r} \|h\|_{L_2(D)} \sum_{i=0}^{T-1} 2^{i/2} + 3 \left( \frac{2r}{\lambda} \right)^{1/2} \sum_{i=T}^{\infty} 2^{i/2} 2^{-\frac{2^i-1}{2N}} \right) .$$

The first sum is $\frac{\sqrt{2}^T - 1}{\sqrt{2} - 1}$. When $T$ is large enough, the second sum is upper bounded by the integral

$$\int_{T-1}^{\infty} 2^{x/2} 2^{-2^x - 1/2N} \, \mathrm{d}x \le \frac{6N}{2^{T/2}} \cdot 2^{-\frac{2^T}{4N}} .$$

To make the form simpler, we bound $\frac{\sqrt{2}^T - 1}{\sqrt{2} - 1}$ by $3 \cdot 2^{T/2}$, and denote $\sup_{h \in \mathcal{H}_r} \|h\|_{L_2(D)}$ by $\rho$. Taking $T$ to be

$$\log_2 \left( 2N \log_2 \left( \frac{1}{\lambda} \right) \right) ,$$

we get the upper bound of the form

$$\mathfrak{R}_D(\mathcal{H}_r) \leq \sqrt{\frac{\ln 16}{m}} \left( 3\rho\sqrt{2N \log_2 \frac{1}{\lambda}} + \frac{18\sqrt{Nr}}{\log_2(1/\lambda)} \right),$$

When $\lambda < 1/2$, $\log_2 1/\lambda > 1$, so we can further enlarge the upper bound to the form

$$\mathfrak{R}_D(\mathcal{H}_r) \leq \sqrt{\frac{(\ln 16)N \log_2 1/\lambda}{m}} \left( 3\sqrt{2}\rho + 18\sqrt{r} \right),$$

$\square$

Next lemma analyzes the expected Rademacher complexity for $\mathcal{H}_r$.

**Lemma 4.** *Assume $\lambda < 1/2$ and $\mathbb{E}h^2(x,y) \leq V\mathbb{E}h(x,y)$. Then*

$$\mathfrak{R}_m(\mathcal{H}_r) \leq C_1\sqrt{\frac{N(V+1)\log_2(1/\lambda)}{m}}\sqrt{r} + C_2\frac{N\log_2(1/\lambda)}{m}.$$

*Proof.* With Lemma 3, we can directly compute the upper bound for $\mathfrak{R}_m(\mathcal{H}_r)$ by taking expectation over $D \sim \mathbb{P}^m$.

$$\mathfrak{R}_m(\mathcal{H}_r) = \mathbb{E}_{D \sim \mathbb{P}^m} \mathfrak{R}_D(\mathcal{H}_r)$$
$$\leq \sqrt{\frac{(\ln 16)N \log_2 1/\lambda}{m}} \left( 3\sqrt{2}\mathbb{E} \sup_{h \in \mathcal{H}_r} \|h\|_{L_2(D)} + 18\sqrt{r} \right).$$

By Jensen's inequality and A.8.5 in Steinwart and Christmann [2008], we have

$$\mathbb{E} \sup_{h \in \mathcal{H}_r} \|h\|_{L_2(D)} \leq \left( \mathbb{E} \sup_{h \in \mathcal{H}_r} \|h\|_{L_2(D)}^2 \right)^{1/2}$$
$$\leq \left( \mathbb{E} \sup_{h \in \mathcal{H}_r} \frac{1}{m} \sum_{i=1}^{m} h^2(x_i, y_i) \right)^{1/2}$$
$$\leq \left( \sigma^2 + 8\mathfrak{R}_m(\mathcal{H}_r) \right)^{1/2},$$

where $\sigma^2 := \mathbb{E}h^2$. When $\sigma^2 > \mathfrak{R}_m(\mathcal{H}_r)$, we have

$$\mathfrak{R}_m(\mathcal{H}_r) \leq \sqrt{\frac{(\ln 16)N \log_2(1/\lambda)}{m}} \left( 9\sqrt{2}\sigma + 18\sqrt{r} \right)$$
$$\leq \sqrt{\frac{(\ln 16)N \log_2(1/\lambda)}{m}} \left( 9\sqrt{2}\sqrt{Vr} + 18\sqrt{r} \right)$$
$$\leq 36\sqrt{\frac{2(\ln 16)N(V+1) \log_2(1/\lambda)}{m}}\sqrt{r}.$$

The second inequality is because $\mathbb{E}h^2 \leq V\mathbb{E}h$ and $\mathbb{E}h \leq r$ for $h \in \mathcal{H}_r$.

When $\sigma^2 \leq \mathfrak{R}_m(\mathcal{H}_r)$, we have

$$\mathfrak{R}_m(\mathcal{H}_r) \leq \sqrt{\frac{(\ln 16)N \log_2(1/\lambda)}{m}} \left( 9\sqrt{2}\sqrt{\mathfrak{R}_m(\mathcal{H}_r)} + 18\sqrt{r} \right)$$
$$\leq 36\sqrt{\frac{(\ln 16)N \log_2(1/\lambda)}{m}}\sqrt{r} + 36^2\frac{(\ln 16)N \log_2(1/\lambda)}{m}.$$

The last inequality can be obtained by dividing the formula into two cases, either $\mathfrak{R}_m(\mathcal{H}_r) < r$ or $\mathfrak{R}_m(\mathcal{H}_r) \geq r$ and then take the sum of the upper bounds of two cases.

Combining all these inequalities, we finally obtain an upper bound

$$\mathfrak{R}_m(\mathcal{H}_r) \leq C_1\sqrt{\frac{(V+1)N \log_2(1/\lambda)}{m}}\sqrt{r} + C_2\frac{N \log_2(1/\lambda)}{m},$$

where $C_1$ and $C_2$ are two absolute constants. $\square$

The last lemma gives the explicit formula of $\varphi_m(r)$. Now we can get the formula for $r$.

**Lemma 5.** *When*

$$r = (900C_1^2 + 120C_2)N(V+1)\frac{\ln(1/\lambda)}{m} + (5B_0 + 72V)\frac{\ln(1/\delta)}{m} \tag{8}$$

*we have*

$$r \geq \max\{30\varphi_m(r), \frac{72V\ln(1/\delta)}{m}, \frac{5B_0\ln(1/\delta)}{m}\}. \tag{9}$$

It can be check by simply plugging $r$ into $\varphi_m(r)$.

## C  Proof of Theorem 1

With Theorem 3 and Lemma 5, we are almost done with the proof of Theorem 1. The only missing part is an upper bound of the approximation error $R_{\mathbb{P},\lambda}^h(f_0) - R^*$. This upper bound has been established in Proposition 1 in Bach [2017]. We rephrase it as below.

**Theorem 4.** *(Proposition 1 of Bach [2017]) Assume that $\phi$ is an optimized feature map and $f$ belongs to the RKHS $\mathcal{F}$ of $\phi$. For $\delta > 0$, when*

$$N \geq 5d(\mu)\log\left(\frac{16d(\mu)}{\delta}\right), \tag{10}$$

*there exists $\beta \in \mathbb{R}^N$ with norm less than 2, such that*

$$\sup_{\|f\|_{\mathcal{F}} \leq 1} \|f - \beta \cdot \phi_N(\cdot)\|_{L^2(\mathcal{X},\mathbb{P})} \leq 2\sqrt{\mu}, \tag{11}$$

*with probability greater than $1 - \delta$.*

Now we prove two simple lemmas connecting the decay rate of $\Sigma$ to the magnitude of $d(\mu)$.

**Lemma 6.** *If $\lambda_i(\Sigma) \leq c_1 i^{-c_2}$, where $c_2 > 1$, we have*

$$d(\mu) \leq \frac{2c_2}{c_2 - 1}\left(\frac{c_1}{\mu}\right)^{1/c_2}, \tag{12}$$

*for $\mu < c_1$.*
*If $\lambda_i(\Sigma) \leq c_3 \exp(-c_4 i^{1/d})$, we have*

$$d(\mu) \leq 5c_4^{-d}\ln^d(c_3/\mu), \tag{13}$$

*for $\mu < c_3 \exp\left(-\left(c_4 \vee \frac{1}{c_4}\right)d^2\right).$*

*Proof.* Both results make use the following observation:

$$d(\mu) = \sum_{i=1}^{\infty}\frac{\lambda_i}{\lambda_i + \mu} \leq m_\mu + \frac{1}{\mu}\sum_{m_\mu+1}^{\infty}\lambda_i, \tag{14}$$

where $m_\mu = \max\{i : \lambda_i \leq \mu\}|$.

When $\lambda_i \leq c_1 i^{-c_2}$, denote $t_\mu = (c_1/\mu)^{1/c_2}$ and then $m_\mu = \lfloor t_\mu \rfloor$. For the tail part,

$$\frac{1}{\mu}\sum_{m_\mu+1}^{\infty}\lambda_i \leq 1 + \frac{1}{\mu}\int_{t_\mu}^{\infty}c_1 x^{-c_2}\,\mathrm{d}x \tag{15}$$

$$\leq 1 + \frac{1}{c_2 - 1}\left(\frac{c_1}{\mu}\right)^{\frac{1}{c_2}}. \tag{16}$$

Combining them together, when $c_1/\mu > 1$, the constant 1 can be absorbed by the second term with a coefficient 2.

When $\lambda_i \leq c_3 \exp(-c_4 i^{1/d})$, denote $t_\mu = \frac{1}{c_4^d} \ln^d \left( \frac{c_3}{\mu} \right)$, and then $m_\mu = \lfloor t_\mu \rfloor$. For the tail part, we need to discuss different situations.

First, if $d = 1$, then we directly have

$$\frac{1}{\mu} \sum_{m_\mu+1}^{\infty} \frac{\lambda_i}{\lambda_i + \mu} \leq \frac{1}{\mu} \left( \mu + \int_{t_\mu}^{\infty} c_3 \exp(-c_4 x) \, \mathrm{d}x \right) \tag{17}$$

$$= 1 + \frac{1}{c_4} . \tag{18}$$

When $\mu < c_3 \exp(-(c_4 \vee \frac{1}{c_4}))$, we can combine these terms into $3t_\mu$.

Second, if $d \geq 2$, when $\mu \leq c_3 \exp(-c_4 e)$, we have that

$$\exp(-c_4 x^{1/d}) \leq \exp\left(-c_4 \frac{t_\mu^{1/d}}{\ln t_\mu} \ln x\right) = x^{-c_4 \frac{t_\mu^{1/d}}{\ln t_\mu}} . \tag{19}$$

Then,

$$\frac{1}{\mu} \sum_{m_\mu+1}^{\infty} \lambda_i \leq 1 + \frac{1}{\mu} \int_{t_\mu}^{\infty} c_3 \exp(-c_4 x^{-1/d}) \, \mathrm{d}x \tag{20}$$

$$\leq 1 + \frac{c_3}{\mu} \int_{t_\mu}^{\infty} x^{-c_4 \frac{t_\mu^{1/d}}{\ln t_\mu}} \tag{21}$$

$$= 1 + \frac{t_\mu}{c_4 \frac{t_\mu^{1/d}}{\ln t_\mu} - 1} . \tag{22}$$

When $c_4 \geq 1$, we may assume that $\mu \leq c_3 \exp(-c_4 d^2)$, and then

$$c_4 \frac{t_\mu^{1/d}}{\ln t_\mu} - 1 \geq \frac{c_4 d^2}{2d \ln d} \geq \frac{4}{3} . \tag{23}$$

So the upper bound has the form $5t_\mu$.

When $c_4 < 1$, we may assume that $\mu \leq c_3 \exp(-d^2/c_4)$, and then

$$c_4 \frac{t_\mu^{1/d}}{\ln t_\mu} - 1 \geq \frac{d^2/c_4}{2d \ln(d/c_4)} \geq \frac{4}{3} . \tag{24}$$

So the upper bound also has the form $5t_\mu$. $\qquad\square$

Now with all these preparation, we can complete our proof of Theorem 1

*Proof.* Under the assumption of Theorem 1, $B_0 = 1$ and $r^* = 0$ in Theorem 3. By Lemma 5, we have

$$r = (900C_1^2 + 120C_2)N(V+1)\frac{\ln(1/\lambda)}{m} + (5 + 72V)\frac{\ln(1/\delta)}{m} . \tag{25}$$

By Theorem 4, we have

$$R_{\mathbb{P},\lambda}^h(f_0) - R^* \leq 2\sqrt{\mu}R + 4R^2 \frac{\lambda}{2} , \tag{26}$$

with probability $1 - \delta$ when $N \geq 5d(\mu) \log \left( \frac{16d(\mu)}{\delta} \right)$.

When the spectrum of $\Sigma$ decays polynomially,

$$d(\mu) \leq \frac{2c_2}{c_2 - 1} \left( \frac{c_1}{\mu} \right)^{1/c_2} . \tag{27}$$

Assume $m > c_1^{-(2+c_2)/(2c_2)}$. By choosing $\mu = c_1 m^{-\frac{2c_2}{2+c_2}} < c_1$ and $\lambda = m^{-c_2/(2+c_2)}$, we have

$$N = 10c_{1,2}m^{\frac{2}{2+c_2}} \left( \ln(32c_{1,2}m^{\frac{2}{2+c_2}}) + \ln(1/\delta) \right) , \tag{28}$$

and

$$R_{\mathbb{P},\lambda}^h(f_{m,N,\lambda}) - R^* \leq \frac{12R}{m^{\frac{c_2}{2+c_2}}} + \frac{12R^2}{m^{\frac{c_2}{2+c_2}}} \tag{29}$$

$$+ 30C_{1,2}c_{1,2}(\ln 32c_{1,2} + \frac{2}{2+c_2}\ln m + \ln(1/\delta))(V+1)\frac{c_2}{2+c_2}\frac{\ln m}{m^{\frac{c_2}{2+c_2}}} \tag{30}$$

$$+ \frac{15 + 216V}{m}\ln(1/\delta)\,, \tag{31}$$

with probability $1 - 4\delta$, where

$$C_{1,2} = 900C_1^2 + 120C_2, \quad c_{1,2} = \frac{c_2 c_1^{1/c_2}}{c_2 - 1}\,. \tag{32}$$

When the spectrum of $\Sigma$ decays sub-exponentially,

$$d(\mu) \leq 5c_4^{-d}\ln^d(c_3/\mu)\,. \tag{33}$$

Assume that $m > \exp(-(c_4 \vee \frac{1}{c_4})d^2/2)$. By choosing $\mu = c_3/m^2$ and $\lambda = 1/m$, we have

$$N = 25c_{d,4}\ln^d(m)(\ln(80c_{d,4}\ln^d(m)) + \ln(1/\delta))\,, \tag{34}$$

and

$$R_{\mathbb{P},\lambda}^h(f_{m,N,\lambda}) - R^* \leq \frac{12R\sqrt{c_3}}{m} \tag{35}$$

$$+ \frac{12R^2}{m} + 150C_{1,2}c_{d,4}(\ln 160c_{d,4} + d\ln\ln m + \ln(1/\delta))(V+1)\frac{\ln^{d+1}m}{m} \tag{36}$$

$$+ \frac{15 + 216V}{m}\ln(1/\delta)\,, \tag{37}$$

with probability $1 - 4\delta$, where

$$C_{1,2} = 900C_1^2 + 120C_2, \quad c_{d,4} = \left(\frac{2}{c_4}\right)^d\,. \tag{38}$$

$\square$

## D   Proof of Theorem 2

Theorem 2 requires a further analysis of the approximation error of RKHS to the Bayes classifier. This part adopts Steinwart and Christmann [2008]'s idea of margin noise exponent. We say that the data distribution $\mathbb{P}$ has margin noise exponent $\beta > 0$ if there exists a positive constant $c$ such that

$$\int_{\{x:\Delta(x)<t\}} |y|\mathrm{d}\mathbb{P}(x,y) \leq ct^{-\beta} \quad \forall t \in (0,1)\,. \tag{39}$$

Therefore, infinite $\beta$ corresponds to our separation condition with $\tau = 1$. However, the original proof of the approximation error that works with the margin noise exponent cannot be generalized to the case of infinite $\beta$, because the coefficient $\Gamma(d + \beta)/2^d$ will blow up (see Theorem 8.18 in Steinwart and Christmann [2008]). This issue can be resolved by modifying the original proof, as shown below.

**Lemma 7.** *Assume that there exists $\tau > 0$ such that*

$$\int_{\{x:\Delta(x)<t\}} |2\eta(x) - 1|\,\mathrm{d}\mathbb{P}_{\mathcal{X}}(x) = 0\,, \forall t < \tau\,, \tag{40}$$

*where $\mathcal{X} \subset B^d(\rho)$ and $\eta(x)$ is a version of $\mathbb{P}(y = 1|x)$. Then there exists a function $f$ in the RKHS generated by the kernel*

$$k_\gamma(x, x') = \exp\left(-\frac{\|x - x'\|^2}{2\gamma^2}\right) \tag{41}$$

*where $\gamma < \tau/\sqrt{d-1}$ such that*

$$R^h(f) - R^* < \frac{4\tau^{d-2}}{\Gamma(d/2)} \exp\left(-\frac{\tau^2}{\gamma^2}\right) \gamma^{d-2},$$

$$\|f\|_{\mathcal{F}} \leq \frac{(\sqrt{\pi/2}\rho^2)^{d/2}}{\Gamma(d/2+1)} \gamma^{-d/2}$$

*and*

$$|f(x)| \leq 1. \tag{42}$$

*Proof.* First we define

$$\mathcal{X}_y := \{x : (2\eta(x) - 1)y > 0\} \text{ for } y = \pm 1, \tag{43}$$

and $g(x) := (\sqrt{2\pi}\gamma)^{-d/2}\text{sign}(2\eta(x) - 1)$. It is square integrable since $\eta(x) = 1/2$ for all $x \notin \mathcal{X}$. Then we map $g$ onto the RKHS by the integral operator determined by $k_\gamma$,

$$f(x) := \int_{\mathbb{R}^d} \phi_\gamma(t; x)g(t) \, \mathrm{d}t, \tag{44}$$

where

$$\phi_\gamma(t; x) = \left(\frac{2}{\pi\gamma^2}\right)^{d/4} \exp\left(-\frac{\|x - t\|^2}{\gamma^2}\right). \tag{45}$$

Note that it is a special property of Gaussian kernel that the feature map onto $L^2(\mathbb{R}^d)$ also has a Gaussian form. For other type of kernels, we may not have such a convenient characterization.

We know that

$$\|f\|_{\mathcal{H}} = \|g\|_{L^2} \leq \frac{\sqrt{\text{Vol}(B^d(\rho))}}{(\sqrt{2\pi}\gamma)^{d/2}} = \frac{(\sqrt{\pi/2}\rho^2)^{d/2}}{\Gamma(d/2+1)} \gamma^{-d/2}. \tag{46}$$

Moreoever,

$$|f(x)| \leq \int_{\mathbb{R}^d} \phi_\gamma(t; x)(\sqrt{2\pi}\gamma)^{-d/2} \, \mathrm{d}t$$

$$= (\pi\gamma^2)^{-d/2} \int_{\mathbb{R}^d} \exp\left(-\frac{\|x - t\|^2}{\gamma^2}\right) \, \mathrm{d}t$$

$$= 1.$$

Since $f$ is uniformly bounded by 1, by Zhang's inequality, we have

$$R^h(f) - R^* = \mathbb{E}_{\mathbb{P}_{\mathcal{X}}}(|f(x) - \text{sign}(2\eta(x) - 1)||2\eta(x) - 1|). \tag{47}$$

Now we give an upper bound on $|f(x) - \text{sign}(2\eta(x) - 1)|$. Assume $x \in \mathcal{X}_1$. Then we know that $f(x) \leq \text{sign}(2\eta(x) - 1) = 1$,

$$1 - f(x) = 1 - \left(\frac{1}{\pi\gamma^2}\right)^{d/2} \int_{\mathbb{R}^d} \exp\left(-\frac{\|x - t\|^2}{\gamma^2}\right) \text{sign}(2\eta(t) - 1) \, \mathrm{d}t$$

$$= 1 - \left(\frac{1}{\pi\gamma^2}\right)^{d/2} \int_{\mathcal{X}_1} \exp\left(-\frac{\|x - t\|^2}{\gamma^2}\right) \, \mathrm{d}t$$

$$+ \left(\frac{1}{\pi\gamma^2}\right)^{d/2} \int_{\mathcal{X}_{-1}} \exp\left(-\frac{\|x - t\|^2}{\gamma^2}\right) \, \mathrm{d}t$$

$$\leq 2 - 2\left(\frac{1}{\pi\gamma^2}\right)^{d/2} \int_{B(x,\Delta(x))} \exp\left(-\frac{\|x - t\|^2}{\gamma^2}\right) \, \mathrm{d}t$$

$$\leq 2 - 2\left(\frac{1}{\pi\gamma^2}\right)^{d/2} \int_{B(0,\Delta(x))} \exp\left(-\frac{\|t\|^2}{\gamma^2}\right) \, \mathrm{d}t$$

$$= 2\left(\frac{1}{\pi\gamma^2}\right)^{d/2} \int_{\mathbb{R}^d \setminus B(0,\Delta(x))} \exp\left(-\frac{\|t\|^2}{\gamma^2}\right) \, \mathrm{d}t$$

$$= \frac{4}{\Gamma(d/2)\gamma^d} \int_{\Delta(x)}^{\infty} \exp\left(-\frac{r^2}{\gamma^2}\right) r^{d-1} \, \mathrm{d}r.$$

Here the key is that $B(x, \Delta(x)) \subset \mathcal{X}_1$ when $x \in \mathcal{X}_1$. For $x \in \mathcal{X}_{-1}$, we have the same upper bound for $1 + f(x)$. Therefore, we have

$$
\begin{aligned}
R^h(f) - R^* &\leq \frac{4}{\Gamma(d/2)\gamma^d} \int_{\mathcal{X}} \int_0^\infty \mathbf{1}_{(\Delta(x),\infty)}(r) \exp\left(-\frac{r^2}{\gamma^2}\right) r^{d-1} |2\eta(x) - 1| \, \mathrm{d}r \mathrm{d}\mathbb{P}_{\mathcal{X}}(x) \\
&= \frac{4}{\Gamma(d/2)\gamma^d} \int_0^\infty \int_{\mathcal{X}} \mathbf{1}_{(0,r)}(\Delta(x)) \exp\left(-\frac{r^2}{\gamma^2}\right) r^{d-1} |2\eta(x) - 1| \, \mathrm{d}\mathbb{P}_{\mathcal{X}}(x)\mathrm{d}r \\
&\leq \frac{4}{\Gamma(d/2)\gamma^d} \int_\tau^\infty \exp\left(-\frac{r^2}{\gamma^2}\right) r^{d-1} \, \mathrm{d}r
\end{aligned}
$$

To get the last line, we apply the assumption on the expected label clarity. Now we only need to give an estimate of the integral.

$$
\int_\tau^\infty \exp\left(-\frac{r^2}{\gamma^2}\right) r^{d-1} \, \mathrm{d}r \leq \int_\tau^\infty C \exp\left(-\alpha\frac{r^2}{\gamma^2}\right) \, \mathrm{d}r \tag{48}
$$

where

$$
C = \tau^{d-1} \exp(-(d-1)/2) \quad \alpha = 1 - 2\gamma^2\tau^{-2}(d-1). \tag{49}
$$

It is required that $\gamma < \sqrt{2}\tau/\sqrt{d-1}$ so that $\alpha > 0$. And then we can give an upper bound to the excess risk

$$
R^h(f) - R^* \leq \frac{4\tau^d}{\Gamma(d/2)(2\tau^2 - (d-1)\gamma^2)} \exp\left(-\frac{\tau^2}{\gamma^2}\right) \gamma^{d-2}. \tag{50}
$$

If we further require that $\gamma < \tau/\sqrt{d-1}$, then we have a simpler upper bound,

$$
\frac{4\tau^{d-2}}{\Gamma(d/2)} \exp\left(-\frac{\tau^2}{\gamma^2}\right) \gamma^{d-2}. \tag{51}
$$

$\square$

Some remarks on this result:

1. The proof follows almost step by step the proof of Steinwart and Christmann [2008]. The only difference occurs at where we apply our assumption.

2. The approximation error is basically dominated by $\exp(-c/\gamma^2)$, and thus leaves us large room for balancing with the norm of the approximator.

3. The proof here only works for Gaussian kernel. A similar conclusion may hold for General RBF kernels using the fact that any RBF kernel can be expressed as an average of Gaussian kernel over different values of $\gamma$. A relevant reference is Scovel et al. [2010].

The last component for the proof of Theorem 2 is the sub-exponential decay rate of the spectrum of $\Sigma$ determined by the Gaussian kernel. The distribution of the spectrum of the convolution operator with respect to a distribution density function $p$ has been studied by Widom [1963]. It shows that the number of eigenvalues of $\Sigma$ greater than $\mu$ is asymptotic to $(2\pi)^{-d}$ times the volume of

$$
\left\{ (x, \xi) : p(x)\hat{k}(\xi) > \mu \right\},
$$

where $\hat{k}$ is the Fourier transform of the kernel function $k$. By applying Widom [1963]'s work in our case, we have the following lemma. It is essentially Corollary 27 in Eric et al. [2008], but our version explicitly shows the dependence on the band width $\beta$.

**Lemma 8.** *Assume $\hat{k}(\xi) \leq \alpha \exp(-\beta\|\xi\|^2)$. If the density function $p(x)$ of probability distribution $\mathbb{P}_{\mathcal{X}}$ is bounded by $B$ and $\mathcal{X}$ is a bounded subset of $\mathbb{R}^d$ with radius $\rho$, then*

$$
\lambda_i(\Sigma) \leq C\alpha B \exp\left(-\beta\left(\frac{4\Gamma^{4/d}(d/2 + 1)}{\pi^{4/d}\rho^2}\right) i^{2/d}\right),
$$

*where $\lambda_1 \geq \lambda_2 \geq \cdots$ are eigenvalues of $\Sigma$ in descending order.*

*Proof.* Denote by $E_t$ the set

$$\left\{(x,\xi) : \hat{k}(\xi)p(x) > t\right\} .$$

The volume, that is, the Lebesgue measure of $E_t$ is denoted by $\mathrm{Vol}(E_t)$. By Theorem II of Widom [1963], the non-increasing function $\phi(\alpha)$ defined on $\mathbb{R}^+$ which is equi-measurable with $p(x)\hat{k}(\xi)$ describes the behaviour of $\lambda_i$s. Indeed, $\lambda_i \leq C\phi((2\pi)^d i)$. By the volume formula of $2d$-dimensional ball we have the following estimate,

$$\sup\{s \in \mathbb{R}^+ : \phi(s) > t\} = \mathrm{Vol}(E_t)$$

$$\leq C_{d,\rho} \left(\frac{\ln(\alpha B/t)}{\beta}\right)^{d/2} ,$$

where

$$C_{d,\rho} = \frac{\rho^d \pi^{d+2}}{\Gamma^2(d/2+1)} . \tag{52}$$

Solving for $t$, we know that

$$\phi(s) \leq \alpha B \exp\left(-\beta\left(\frac{s}{A}\right)^{2/d}\right) .$$

Therefore, we have

$$\lambda_i(\Sigma) \leq C\alpha B \exp\left(-\beta\left(\frac{(2\pi)^d i}{A}\right)^{2/d}\right) \tag{53}$$

$$= C\alpha B \exp\left(-\beta\left(\frac{4\Gamma^{4/d}(d/2+1)}{\pi^{4/d}\rho^2}\right)i^{2/d}\right) . \tag{54}$$

$\square$

Now we can prove Theorem 2.

*Proof.* Note that, by Lemma 7, we can construct $g \in \mathcal{F}$ such that $R_{\mathbb{P},\lambda}^h - R^*$ is controlled. And by Theorem 4, we can find an $f_0 \in \mathcal{F}_N$ with similar risk to $g$. And this will be our $f_0$ as required by Theorem 3. So we have

$$R_{\mathbb{P},\lambda}^h(f_0) - R^* \leq \frac{2(\sqrt{\pi/2}\rho^2)^d}{\Gamma^2(d/2+1)}\frac{\lambda}{\gamma^d} + \frac{2(\sqrt{\pi/2}\rho^2)^{d/2}}{\Gamma(d/2+1)}\sqrt{\mu} \tag{55}$$

$$+ \frac{4\tau^{d-2}}{\Gamma(d/2)}\exp\left(\frac{\tau^2}{\gamma^2}\right)\gamma^{d-2} , \tag{56}$$

and $\|f_0\|_{\mathcal{F}_N} \leq 2$, with probability $1-\delta$, when $N = 5d(\mu)\ln(16d(\mu)/\delta)$. We choose $\gamma = \tau/\sqrt{\ln m}$ and $\lambda = 1/m$. Under the boundedness assumption on the density function and the property of Gaussian kernel, we know that by Lemma 8,

$$\lambda_i(\Sigma) \leq C\gamma B \exp\left(-\gamma^2\frac{4\Gamma^{4/d}(d/2+1)}{\pi^{4/d}\rho^2}i^{2/d}\right) . \tag{57}$$

And similar to the second part of Theorem 1, by identifying

$$c_3 = C\gamma B = CB\tau/\sqrt{\ln m} \quad c_4 = \frac{4\tau^2\Gamma^{4/d}(d/2+1)}{\pi^{4/d}\rho^2 \ln m} := \frac{A}{\ln m} , \tag{58}$$

and choosing $\mu = c_3/(m^{2d^2} \vee \exp(\frac{d^2}{c_4} \vee c_4 d^2))$, we have

$$d(\mu) \leq 5d^{2d}(c_4^{-2d} \vee 1 \vee c_4^{-d}2^d \ln^d m) . \tag{59}$$

Then when $m \geq \exp(A)$, we have $d(\mu) \leq 5(A^2 \wedge A/2)^{-d}\ln^{2d} m$, and

$$N = 5d(\mu)(\ln(16d(\mu)) + \ln(1/\delta)) \tag{60}$$

$$\leq 25(A^2 \wedge A/2)^{-d}\ln^{2d} m(\ln(80(A^2 \wedge A/2)^{-d}) + 2d\ln\ln m + \ln(1/\delta)) . \tag{61}$$

Plug $N$ and $\lambda$ into Equation 8.

$$3r = 75C_{1,2}c_{d,\tau,\rho}(\ln(80c_{d,\tau,\rho}) + 2d\ln\ln m + \ln(1/\delta))(V+1)\frac{\ln^{2d+1}m}{m} \tag{62}$$

$$+ \frac{15 + 216V}{m}\ln(1/\delta) + 3r^*\,, \tag{63}$$

where

$$C_{1,2} = 900C_1^2 + 120C_2, \quad c_d = (A^2 \wedge A/2)^{-d}\,. \tag{64}$$

We can bound $r^*$ by $R_{\mathbb{P},\lambda}^h(f_0) - R^*$. Therefore, the overall upper bound on the excess error is

$$R_{\mathbb{P},\lambda}^1(f_{m,N,\lambda}) - R^* \leq \frac{18(\sqrt{\pi/2}\rho^2)^d}{\Gamma^2(d/2+1)}\frac{\ln^{d/2}m}{\tau m} + \frac{18(\sqrt{\pi/2}\rho^2)^{d/2}}{\Gamma(d/2+1)}\frac{\sqrt{CB\tau}\ln^{1/4}m}{m^{d2}} \tag{65}$$

$$+ \frac{36\tau^{d-2}}{\Gamma(d/2)}\frac{\tau^{d-2}}{m\ln^{d/2-1}m} \tag{66}$$

$$+ 75C_{1,2}c_{d,\tau,\rho}(\ln(80c_{d,\tau,\rho}) + 2d\ln\ln m + \ln(1/\delta))(V+1)\frac{\ln^{2d+1}m}{m} \tag{67}$$

$$+ \frac{15 + 216V}{m}\ln(1/\delta)\,. \tag{68}$$

$\square$

## E  Learning Rate without Optimized Feature Maps

In this section, we discuss the learning rate of RFSVM without an optimized feature map. As shown by Rudi and Rosasco [2017], RFKRR can achieve excess risk of $O(1/\sqrt{m})$ using $O(\sqrt{m}\log(m))$ features. However, it is inappropriate to directly compare this result with the learning rate in classification scenario. Because as surrogate loss functions, least square loss has a different calibration function with for example hinge loss. Basically, $O(\epsilon)$ risk under square loss only implies $O(\sqrt{\epsilon})$ risk under $0-1$ loss, while $O(\epsilon)$ risk under hinge loss implies $O(\epsilon)$ risk under $0-1$ loss. Therefore, Rudi and Rosasco [2017]'s analysis only implies an excess risk of $O(m^{-1/4})$ in classification problems with $\tilde{O}(\sqrt{m})$ features.

For RFSVM, we expect a similar result. Without assuming an optimized feature map, the leverage score can only be upper bounded by $\kappa^2/\mu$, where $\kappa$ is the upper bound on the function $\phi(\omega; x)$ for all $\omega, x$. Substituting $\kappa^2/\mu$ for $d(\mu)$ in the proofs of learning rates, we need to balance $\sqrt{\mu}$ with $1/(\mu m)$ to achieve the optimal rate. This balance is not affected by the spectrum of $\Sigma$ or whether $f_{\mathbb{P}}^*$ belongs to $\mathcal{F}$. Obviously, setting $\mu = m^{-2/3}$, we get a learning rate of $m^{-1/3}$, with $\tilde{O}(m^{2/3})$ random features. Even though this result is also new for RFSVM in regularized formulation, the gap to previous analysis like Rahimi and Recht [2008] is too large. Considering that the random features used in practice that are not optimized also have quite good performance, we need further analysis on RFSVM without optimized feature map.

# F    Supplementary Figures

Figure 5: Comparison between RFSVMs with KSVM Using Gaussian Kernel.

"ksvm" is for KSVM with Gaussian kernel, "unif" is for RFSVM with direct feature sampling, and "opt" is for RFSVM with reweighted feature sampling. Error bars represent standard deviation over 10 runs. Each sub-figure shows the performance of RFSVM with different number of features $N$.

Figure 6: The excess risks of RFSVMs with the simple random feature selection ("unif") and the reweighted feature selection ("opt") are shown for different sample sizes in the binary classification task over 10 dimensional data. The data with probability 0.9 to be -1 are within the 10 dimensional ball centered at the origin and radius 0.9, and the data with probability 0.9 to be 1 are within the shell of radius 1.1 to 2. The error rate is the excess risk. The error bars represent the standard deviation over 10 runs.

Figure 7: The classification accuracy of RFSVM with the simple random feature selection ("unif") and the reweighted feature selection ("opt") are shown for different sample sizes in the hand-written digit recognition (MNIST)