[Reviews · NeurIPS 2018]

Reviewer 1



The authors analyze the use of random Fourier Features (RFF) for training linear support vector machines in the RFF feature space. Their result bounds the expected risk difference of the hinge loss (i.e. the generalization error on unseen data) for a number of different scenarios. The analysis is based on a number of assumptions, on the noise distribution, on the RKHS containing the optimal classifier, and most crucially on access to the optimal feature weights by Bach 2017. The first main result is a fast rate for kernels whose spectrum decays polynomially when the Bayes classifier is in the feature space. The second result is a fast rate specifically for the Gaussian kernel when the Bayes classifier is not necessarily in the feature space, but the data is separable by some minimum distance. Since the optimal feature weights are not available in practice, the authors rely on Bach's algorithm to approximate them in their experiments. As in non-parametric regression, optimized feature weights are beneficial for the performance of the RFF SVM. I would like to congratulate the authors, this is an excellent contribution to a worthwhile problem, and a well structured and well written paper. I have a few comments on technical aspects, presentation and experiments below. Apart from that, I am happy for the paper to get accepted. As a side note: as with most theory papers, the conference format is a bit unfortunate, as the main paper merely leaves space to state the results. I would prefer to see these articles in ML journals with a thorough review process that covers all proofs in detail (which is quite hard to do for 6 papers during NIPS reviews). Significance. This is the first analysis of the generalization error using RFF and SVM, and as the authors point out, most of the previous analysis for RFF were based on the idea of perturbing the optimization problem and did not lead to error bounds that justified the empirical success of RFF. What is nice that the authors treat the RFF SVM as a seperate learning model, rather than a perturbed version of the kernel SVM. What is also nice is that the rates are actually fast enough (for low dimensions) to justify the empirical success of RFF SVMs that use a fewer random features than data samples. On the downside, the rates are unattainable in practice due to the intractability of the optimized feature weights, and the rates for general uniform features are not satisfactory. A second downside is that the results for the mis-specified case (as fast as they are!) are only for the Gaussian kernel. There are approaches in regression like settings, where the assumption that the data generating function does not lie in the RKHS is relaxed using interpolation spaces, e.g. Theorem 12 in [R1]. [R1] "Density Estimation in Infinite Dimensional Exponential Families", Sriperumbudur et al, https://arxiv.org/abs/1312.3516 A. Assumptions. A1. The assumption on the data noise seems fairly standard. It would be great, however, if the authors could provide guidelines for practicioners on diagnosing when the low noise assumption is (approximately) valid and when it is not. Such diagnosis statistics are quite common in the statistics community, and they are usually based on post-processing of some kind. A2. It is unfortunate that the results provided are only for the case of having access to the optimized feature weights, and it is even more unfortunate that using uniform features does not seem to result in fast rates in general. Of course the authors make this very clear, and Appendix E provides some additional intuition for using the uniform feature maps (some of it should be in the main text). It is unclear to me what is meant by "general feature maps" in line 237? And what is the conjecture based on? This seems a bit vague as for example Theorem 2 also managed to work around the crucial assumption that the Bayes classifier lies in the RKHS using a specialization of the used kernel. A3. The authors are correct that analysing the reweighted feature selection algorithm by Bach 2017 is out of scope for this paper. But they will have to acknowledge, however, that this "missing link" clearly lessens the significance of the results. A quantification of the error of the approximately optimal reweighting might be combined with the presented results into an error analysis that would be extremely relevant for practicioners. B. Comparison to the Nystrom SVM. The authors mention that they "do not find any previous work on the performance of SVM with Nystrom method on classification problems" (line 241). I presume this is regarding "theoretical analysis of the generalization error", as there is a number of papers that deals with the very matter of RFF SVMs, e.g. [R2, R3]. What I think is missing in the paper is an acknowledgement (or a reproduction) of previous experiments that were done to compare the performance of Nystrom and RFF SVMs, e.g. [R3]. This would be in particular useful to determine which method works better in which situation, or at least to contrast the two approaches from a practical perspective. These would just one or two more lines in the plots. [R2] "Scaling up kernel svm on limited resources: A low-rank linearization approach", Zhang et al, http://proceedings.mlr.press/v22/zhang12d/zhang12d.pdf [R3] "Nystrom Method vs Random Fourier Features: A Theoretical and Empirical Comparison", Yang et al, https://papers.nips.cc/paper/4588-nystrom-method-vs-random-fourier-features-a-theoretical-and-empirical-comparison.pdf C. High dimensional data. It is only very briefly mentioned at the end that experiments on higher-dimensional data are a bit disappointing. Placing these experiments in the Appendix is not optimal, not just because they are the only experiments on non-synthetic data. There is an interesting question here: is it the approximate feature selection algorithm or the feature weighting itself? D. Proof outline I think an overview of the proof strategy and significance is a must for the main paper. The proof relies on some quite heavy external results, namely Theorem 3 and 4, or modified external results (Lemma 7), and it would be great to get a better feeling for what parts of the proof contain the most significant original steps. There is a number of smaller Lemmas that seem more or less straight-forward involving taking expectations and simple bounding, but there are also some heavier parts and it would be great if the authors could shed some light onto those. I would like to thank the authors for the nice response.

Reviewer 2



The authors prove that under low noise assumptions, the support vector machine with a few random features (RFSVM) (fewer than the sample size) can achieve fast learning rate, extending the previous fast rate analysis of random features method from least square loss to 0-1 loss. The results are interesting, assuming that Assumption 2 is reasonable and the proof is correct. The presentation of the paper needs to be polished a bit. I only have one major comment. Is there any practical example for Assumption 2. What will Theorem 1 be without Assumption 2? ==After rebuttal==== I am a bit disappointed that the authors did not manage to address my comment about Assumption 2. The first part of Assumption 2 (on the random features) is less common, although it has been used in the literature. I kept my evaluation unchanged and I agree that the paper should be accepted as there are some interesting results. But more elaboration on Assumption 2, or deriving results without Assumption 2 would make the paper a great one.

Reviewer 3



This paper deals with the analysis of the kernel random feature scheme for linear SVM. In line with the recent work by Bach on optimal sampling scheme for random features and by Rudi and Rosasco on kernel random feature regression, the authors prove that, under certain hypotheses, learning was feasible with a number of features that is actually lower than the sample size. I thus consider this work to be important, since despite their success due to their simplicity it is still not clear in which situation using random features brings an actual theoretical advantage. While I very much liked this paper, there is still a bit of room for improvement. First of all, the main definition of the paper, which is the optimal sampling scheme, is quite confusing. Who is mu ? Is its existence assumed or always guaranteed ? Similarly, although their meaning is clear the lambda(Sigma) have not properly been defined. I would maybe displace everything that concerns optimized sampling scheme (including the discussion on limitations and the approximate algorithm by Bach) to a dedicated subsection, since the optimal sampling scheme and the practical algorithm that goes with it constitutes for me the crucial point of the paper. Similarly, I would mention it more clearly in the intro. Secondly, I would elaborate a bit more on the role of the dimension, even though the authors already do that. In particular, while reading the paper the reader has the impression that the "sub-exponential" case is always better than the polynomial case, unless I'm wrong in high dimension this is not the case and the number of features may be worse in the second case. This should be stated clearly. Although, is it possible to give a few examples of usual Sigma and the decay of their eigenvalues after the definition of the decays? If space permits, I would include a small sketch of proof in the body of the paper, to outline the novel parts of it. Finally, a few typos and comments: - l94-95: I believe the variance for the RF sampling is gamma^{-2} for the gaussian kernel (inverted compared to the kernel) - l128-129: "we consider a simpler hypothesis class only by forcing..." -> not clear - l142: gaurantees - l185: "to remove" -> removing - App F: Supplimentary In conclusion, I believe this work to be an important step forward in the still popular random feature scheme. Although the results have some limitations due to optimal sampling scheme and possible exponential dependence in the dimension that could be stated more clearly, the overall paper is easy to read and well-explained. --- After feedback --- I thank the authors for answering most of the points I raised. I understand their choice to keep the structure of the paper as is. A strong accept.